# The Epidemiological Analysis of COVID-19 Outbreaks in Nursing Homes during the Period of Omicron Variant Predominance

**DOI:** 10.3390/healthcare11212868

**Published:** 2023-10-31

**Authors:** Jeffrey Che-Hung Tsai, Ying-Ying Chang, Chiann-Yi Hsu, Hui-Ju Chen, Feng-Tse Chan, Zhi-Yuan Shi

**Affiliations:** 1Department of Emergency Medicine, Taichung Veterans General Hospital, Puli Branch, Nantou 545402, Taiwan; erdr2181@gmail.com; 2School of Medicine, College of Medicine, National Yang Ming Chiao Tung University, Taipei 112304, Taiwan; 3College of Medicine, National Chung Hsing University, Taichung 402202, Taiwan; 4Nursing Department, Taichung Veterans General Hospital, Puli Branch, Nantou 545402, Taiwan; 5Biostatistics Group, Department of Medical Research, Taichung Veterans General Hospital, Taichung 407219, Taiwan; 6Infection Control Team, Taichung Veterans General Hospital, Puli Branch, Nantou 545402, Taiwan; 7Division of Infectious Diseases, Department of Internal Medicine, Taichung Veterans General Hospital, Taichung 407219, Taiwan; 8Department of Industrial Engineering & Enterprise Information, Tunghai University, Taichung 407224, Taiwan

**Keywords:** COVID-19, Omicron, outbreak, nursing home, risk factors, Barthel index

## Abstract

Background: The main purpose of this study was to evaluate the epidemic trend and risk factors associated with COVID-19 outbreaks in nursing homes during the period of Omicron variant predominance. Methods: The study analyzed the risk factors associated with SARS-CoV-2 infection and death among the 327 residents and 129 healthcare workers (HCWs) in three hospital-affiliated nursing homes through a multivariate Cox regression model. Results: The rates of receiving a COVID-19 booster dose were 70.3% for the residents and 93.0% for the healthcare workers (HCWs), respectively. A number of asymptomatic individuals, including 54 (16.5%) residents and 15 (11.6%) HCWs, were detected through mass screening surveillance tests. The COVID-19 infection rates during the outbreaks were 41.6% among residents and 48.1% among HCWs, respectively. The case fatality rate among residents was 10.3%. None of the HCWs were hospitalized or died. The multivariate Cox regression model showed that the risk of COVID-19 infection increased in males (HR 2.46; 95% CI 1.47–4.11; *p* = 0.001), Barthel index ≥ 61 (HR 1.93; 95% CI 1.18–3.17; *p* = 0.009), and dementia (HR 1.61; 95% CI 1.14–2.27; *p* = 0.007). The risk of COVID-19 death increased with pneumonia (HR 11.03; 95% CI 3.02–40.31; *p* < 0.001), hospitalization (HR 7.18; 95% CI 1.97–26.25; *p* = 0.003), and admission to an intensive care unit (HR 8.67; 95% CI 2.79–26.89; *p* < 0.001). Conclusions: This study highlighted the high infection rates with a substantial proportion of asymptomatic infections for both residents and HCWs, as well as a high case fatality rate for the residents among nursing homes during the Omicron epidemic period. We suggest implementing mass screening through regular surveillance testing as an effective strategy for early detection of COVID-19 and for preventing transmission during an epidemic period. Pneumonia is the primary risk associated with COVID-19 death. Early detection and prompt treatment of pneumonia for vulnerable residents in nursing homes are crucial to protect them from potential mortality.

## 1. Introduction

Coronavirus disease 2019 (COVID-19) was first discovered in Wuhan City, China, in December 2019, and it quickly spread worldwide. There have been about 770 million confirmed cases of COVID-19, including 6.9 million deaths, as reported by the World Health Organization (WHO) in September 2023 [1]. Four variants of concern, namely Alpha (B.1.1.7), Beta (B.1.351), Gamma (P.1), and Delta (B.1.617.2), caused the pandemic between December 2020 and May 2021 [2]. The Omicron variant (B.1.1.529) was first discovered in South Africa in November 2021 [3]. The Omicron variant has replaced the Delta variant and become the dominant strain around the world [4].

Residents in long-term care facilities (LTCFs) are mostly elderly, with many comorbidities and a higher degree of disability. Therefore, they are vulnerable and have high risks of severe illness and death from COVID-19. Most residents live in clustered spaces with shared rooms, bathrooms, and toilets, which have been associated with COVID-19 outbreaks [5]. Nursing homes worldwide have extensively suffered from the COVID-19 pandemic [6]. In an analysis of pooled data from 14 countries across the USA and Europe up to October 2021, the proportion of COVID-19 cases among the occupied beds in nursing homes ranged from 2.2% in Finland, 12% in Germany, and 36% in France to 50% in the USA. Although long-term care residents comprise a relatively small percentage of the population, their deaths have consistently accounted for about 40% of the global COVID-19 deaths [6]. The proportion of COVID-19 deaths in nursing homes among all COVID-19 deaths in the total population accounted for 11% in the Czech Republic, 18% in the USA, 24% in Germany, 26% in France, and 50% in Belgium [6]. Perhaps due to its milder clinical presentation, the prevalence of Omicron variant infection and its associated mortality rates since 2022 are currently unclear [7,8].

The risk factors for COVID-19 death have been identified, including old age, male sex, chronic obstructive pulmonary disease, pneumonia, cardiovascular disease, diabetes mellitus, obesity, dementia, frailty, cancer, and end-organ disease [9,10]. The studies for these risk factors have some limitations: First, due to the increasing prevalence of comorbidities with age, the lack of adjustment for confounding in these studies may lead to an overestimation of the influence of each risk factor [10]. Secondly, they are hospital-based studies including patients with different disease severities in the pre-Omicron period [11]. Lastly, studies investigating both the risk factors and ADL associated with COVID-19 infection in nursing homes during the Omicron period are rare. Therefore, there is a need for epidemiologic studies in nursing homes during the Omicron period.

Healthcare workers (HCWs) caring for COVID-19 patients in hospitals or nursing homes are at a higher risk for SARS-CoV-2 transmission and infection [12,13]. According to case-based surveillance data of COVID-19 reported to the European Surveillance System from 31 January 2020 to 13 January 2021, the reported rates for 261,080 HCWs were 10.1% infection, 0.4% hospitalization, 0.03% admission to ICU, and 0.01% death [14]. As the COVID-19 pandemic has continued for more than three years, rare articles have described the trend of infection statistics among HCWs throughout the pandemic course [15]. The issue regarding COVID-19 infection statistics among HCWs needs to be further addressed.

The three nursing homes (two general nursing homes and one psychiatric nursing home) affiliated with Taichung Veteran General Hospital, Puli Branch, have been following the updated infection prevention and control (IPC) measures recommended by the Centers for Disease Control and Prevention during the COVID-19 pandemic [16]. These IPC measures include using proper PPE (a properly fitted N95 respirator, eye protection, gloves, and gown), keeping distance when higher-risk activities are necessary, receiving COVID-19 vaccination, restricting group activities, restricting visitors, mass screening via regular testing for COVID-19 for residents and staff, monitoring for common symptoms of COVID-19, environmental cleaning, isolating and cohorting patients with suspected or confirmed COVID-19, and reporting to infection control practitioners and the Taiwan Centers for Disease Control if an individual tests positive. Although our nursing homes have suspended group activities during the pandemic, non-COVID-19 residents in nursing homes are allowed to dine together in public dining rooms to reduce their psychological stress [17,18]. The conflict between the benefits of group activities and the violation of infection control practices needs to be further addressed [18]. Mass screening through the routine testing of residents and staff, regardless of symptoms, may help reduce the number of infections [19,20,21]. Mass screening may reduce the probability of outbreaks, but the evidence is very unclear [19].

The Omicron variant, BA. 2.3.7 has emerged as the predominant strain and has led to outbreaks in communities and healthcare systems throughout Taiwan since April 2022 [22]. The main purposes of this study are to (1) evaluate the epidemic trend and the variations in characteristics of COVID-19 outbreaks among the residents and HCWs in the three studied nursing homes and (2) analyze the risk factors associated with COVID-19 infection and mortality during the period of Omicron variant predominance.

## 2. Methods

### 2.1. Study Design

This was an observational, retrospective cohort study conducted in the context of the ongoing COVID-19 outbreaks in the three nursing homes affiliated with Taichung Veteran General Hospital, Puli Branch, Taiwan. The study evaluated the epidemic trend and the variations in characteristics among the residents and HCWs in the three nursing homes, as well as the risk factors associated with SARS-CoV-2 infection and related mortality.

### 2.2. Setting

Taichung Veteran General Hospital, Puli Branch, is a suburban district hospital with 328 acute and chronic care beds, serving people in Puli Township and its neighborhood areas. The hospital has three affiliated nursing homes (Unit B, Unit F, and Unit P) located in three separate buildings on the hospital campus, serving both veterans and general civilians. The two general nursing homes (Unit B and Unit F) can accommodate up to 290 general residents. The third psychiatric nursing home (Unit P) can accommodate up to 57 residents with psychiatric diseases. Notably, the HCWs do not work between different units.

The first outbreak case occurred on 16 May 2022, and the last new case was reported on 23 July 2022. The final death related to the outbreak was recorded on 13 August 2022. Therefore, data collection for this outbreak spanned from 5 May 2022 to 13 August 2022.

### 2.3. Study Population

The study population included the residents and HCWs in the three nursing homes. The long-stay residents who had stayed in the nursing homes for more than 7 days before the start date of the outbreaks were included for analysis. The HCWs included registered nurses, nursing assistants, social workers, clerks, housekeepers, and porters working in the three nursing homes. There were 327 residents in total, including 87, 184, and 56 residents in Units B, F, and P, respectively. There were 129 HCWs in total, including 38, 66, and 25 HCWs in Units B, F, and P, respectively.

### 2.4. Epidemiological Survey

When the outbreak happened, mass screening for SARS-CoV-2 infection using rapid antigen tests was performed on the first day (16 May 2022) and then every two days for one week after the index case was found in the nursing homes. COVID-19 was confirmed through a positive SARS-CoV-2 PCR test. Thereafter, mass screening with rapid antigen tests was conducted twice a week for residents and once a week for HCWs from 20 June 2022, following the revised recommendation by the Taiwan Centers for Disease Control.

### 2.5. Study Variables

The risk variables included demographic characteristics (age and sex); the presence of various conditions, including consent to a Do Not Resuscitate order (DNR); medical supports (tube feeding, artificial airway); underlying diseases (hypertension, cerebrovascular accident, dementia, diabetes mellitus, congestive heart failure, coronary artery disease, chronic liver disease, chronic kidney disease, chronic pulmonary disease, end-stage renal disease, advanced malignancy, hyperlipidemia); physical dependency level in ADL measured with the Barthel Index; and booster vaccination rates. All of them were categorical variables, except for age, which was a continuous variable. All these risk variables were analyzed to assess their differences among the three nursing homes.

The main outcome measures were SARS-CoV-2 infection and COVID-19-related death. All the risk variables were analyzed for their association with these two main outcomes using a Cox regression model, while the additional variables, such as pneumonia, hospitalization, and intensive care, were also examined for their relation to COVID-19-related death.

A high proportion of asymptomatic or pre-symptomatic infections among residents and HCWs was a confounding factor. Mass screening using antigen rapid tests or polymerase chain reaction (PCR) tests is an effective strategy for identifying these cases, providing a more accurate estimation of COVID-19 infection rates and associated case fatality rates [20,23].

### 2.6. Data Sources and Measurement

The building space of the three nursing homes was obtained from the hospital administration office. The underlying diseases and COVID-19-related conditions of the residents were retrieved from the electronic medical records of this hospital. The acute and chronic diseases of nursing home residents were tracked and treated at this hospital; thus, we were able to obtain detailed medical records. The demographic data, Barthel index, and vaccination data among the residents were obtained from the nursing records of the nursing homes. The information involving the employee types, demographic data, underlying diseases, and COVID-19-related conditions among the HCWs was collected through a questionnaire administered by the infection control nurse at the hospital. The incidence data of Puli Township were obtained from the public website of the National Center for High-Performance Computing [24].

For data validation, we double-checked the data entries for typographical errors and compared the prevalence rates of the underlying diseases with previously published data. For example, the prevalence rates of hypertension, dementia, diabetes mellitus, chronic obstructive pulmonary disease, and hyperlipidemia were similar to the previously published data worldwide [25,26,27,28,29]. We also checked for internal consistency by comparing related variables to ensure they were logically coherent. For example, although some residents in the psychiatric nursing home had total dependency and severe dependency, they were indeed brought together for meals in the dining room to prevent psychosocial stress.

The Barthel index included the following 10 items: feeding, bathing, grooming, dressing, toilet use, bowels, bladder, mobility, transfers, and climbing stairs. Each item was scored with 0, 5, 10, or 15 points. The summed scores of the Barthel index were categorized as follows: a score of 100 indicated total independency, scores of 91–99 indicated mild dependency, scores of 61 to 90 indicated moderate dependency, scores of 21 to 60 indicated severe dependency, and scores of 0 to 20 indicated total dependency [30].

The vaccination rate is defined as the percentage of 327 individuals who receive a booster dose after completing a 2-dose primary vaccine series, with the booster dose administered ≥14 days before the onset of COVID-19 or before the outbreak for non-infected individuals [31].

The average number of days between the date of a booster dose and the date of SARS-CoV-2 infection was calculated for both residents and HCWs with COVID-19.

The case fatality rate was defined as the total number of COVID-19 deaths divided by the total number of confirmed COVID-19 cases. The COVID-19-related mortality rate was defined as the number of COVID-19-related deaths divided by the total number of residents, including those with and without COVID-19 diagnosis.

The cumulative incidence is defined as dividing the number of new cases by the total population at risk within the study observation period, then multiplying it by 100% to express it as a percentage. The formula is cumulative incidence = (number of new cases/total population at risk within the study observation period) × 100%. The cumulative cases refer to the total number of cases recorded over the study observation period.

The graphs of cumulative incidences and cumulative cases were plotted over the outbreak period to monitor the peaks and trends of COVID-19. A graph was plotted to compare the cumulative incidence between the epidemic in Puli Township and that in the three nursing homes. A graph was plotted to demonstrate the cumulative cases of COVID-19 in the three nursing homes. An epidemic histogram (bar chart) was plotted to show the number of new cases per week in the three nursing homes.

### 2.7. Bias

The prevalence of coronary artery disease may be underestimated due to the lack of invasive diagnostic procedures, such as coronary catheterization, for confirmation. The sample size of the residents for some underlying diseases, such as advanced malignancy, end-stage renal disease, and chronic liver disease, was small and could affect the reliability of the results. The study population was predominantly composed of males, indicating a bias toward males.

### 2.8. Statistical Analysis

In the first step, we used the Chi-square test or Fisher’s exact test to assess the differences in categorical risk variables among the three nursing homes. Fisher’s exact test was specifically chosen for cases with small sample sizes where expected cell counts were less than 5. Additionally, we utilized the ANOVA test to compare continuous variables.

In the next step, all residents were considered as a single study population to analyze the association between risk factors and main outcomes, which included SARS-CoV-2 infection and COVID-19-related mortality. This approach was taken because there were no significant differences observed in the underlying diseases of the residents in the three nursing homes, and the numbers of residents with some underlying diseases were small. The analysis was conducted using a Cox regression model, assessing hazard ratios (HR) along with 95% confidence intervals (CI). A *p*-value of <0.05 was deemed statistically significant. We conducted univariate and multivariate statistical analyses of the variables. Variables demonstrating a statistically significant difference in univariate Cox regression are presented in the results section; they were subjected to multivariate Cox regression analysis. Age was considered to be a major confounding factor affecting the relationship between significant factors and COVID-19-related deaths. Consequently, the statistically significant factors were adjusted by age, the continuous covariate, in the multivariate Cox regression model.

The analyses were performed using the Statistical Package for the Social Science (IBM SPSS version 22.0; International Business Machines Corp, New York, NY, USA).

## 3. Results

### 3.1. Descriptive Data

The building spaces of the three nursing homes are shown in Table 1. The average space per resident was 37.8, 29.8, and 42.4 square meters in Unit B, F, and P, respectively. Additionally, the average dining area space per resident was 30.4, 32.4, and 4.1 square meters in Units B, F, and P, respectively (see Table 1).

The demographics and underlying diseases of the residents in the three nursing homes are listed in Table 2. There were 87, 184, and 56 residents in Unit B, F, and P, respectively. The accommodated residents in the three nursing homes had variable patterns of characteristics. Most (79.8%) of the residents in all three units were male, and the percentage of male residents was higher in Unit B (89.7%). A lower percentage of residents in Unit P had consented to a DNR order (16.1% in Unit P, 59.8% in Unit B, and 59.8% in Unit F, *p* < 0.001).

Residents in Unit B appeared to have a higher percentage of tube feeding (65.5% in Unit B, 44% in Unit F, and 0% in Unit P, *p* < 0.001). Residents in Unit B appeared to have a higher percentage of artificial airways (11.5% in Unit B, 2.2% in Unit F, and 0% in Unit P, *p* < 0.001). None of the residents in Unit P required either tube feeding or an artificial airway. The dependency patterns of the three nursing homes showed a significant difference (*p* < 0.001). Most residents in Unit B (78.2%) and Unit F (70.7%) had total dependency, while residents in Unit P had a lower rate (8.9%) of total dependency (*p* < 0.001).

About 33.3% of all residents had diabetes mellitus, and 41.6% of residents had cerebrovascular diseases. Unit P had more residents with dementia (73.2% in Unit P, 37.9% in Unit B, and 34.2% in Unit F, *p* < 0.001) and fewer residents with coronary artery disease than other units (3.6% in Unit P, 19.5% in Unit B, and 10.9% in Unit F, *p* = 0.013) (Table 2).

A total of 101 HCWs responded to the questionnaire. The demographic characteristics and underlying diseases among the HCWs are listed in Table 3. There were 28 registered nurses, 59 nursing assistants, three social workers, seven housekeepers and porters, and four clerks. Females accounted for 96.4% of registered nurses and 84.7% of the nursing assistants. The average age among the 101 HCWs was 42.3 ± 11.9. The majority of the HCWs were healthy, while 15.8% had hypertension and 6.9% had diabetes mellitus.

### 3.2. Outcome Data

Vaccination- and COVID-19-related statistics among the residents and HCWs in the three nursing homes are shown in Table 4. The overall vaccination rate of a booster dose was 70.3% for residents. The average intervals between a booster dose and the occurrence of infection were 80.9 ± 29.1, 97.7 ± 26.1, and 134.5 ± 22.8 days for the residents in Units B, F, and P, respectively.

A total of 54 (16.5%) asymptomatic residents were detected through mass screening surveillance tests. The overall infection rate was 41.6% (136/327) among all residents. Among the three nursing homes, the residents in the psychiatric nursing home had a significantly higher vaccination rate (89.3%, *p* = 0.001) and infection rate (91.1%, *p* < 0.001).

About 13.5% (44/327) of all residents were hospitalized. The residents in Unit B were more likely to be hospitalized when they were infected (59.1% in Unit B, 28.6% in Unit F, and 25.5% in Unit P, *p* = 0.013). The overall case fatality rate among the infected residents was 10.3% (14/136), and the COVID-19-related mortality was 4.3%. The residents in Unit P seemed to have a lower case fatality rate (5.9% in Unit P, 9.1% in Unit B, and 14.3% in Unit F), but there was no statistically significant difference (*p* = 0.355).

The overall vaccination rate of a booster dose was 93.0% for HCWs. The average intervals between a booster dose and the occurrence of infection were 110.5 ± 35.6, 114.8 ± 61.6, and 125.7 ± 18.3 days for the HCWs in Units B, F, and P, respectively.

A total of 15 (11.6%) asymptomatic HCWs were detected through mass screening surveillance tests. The overall infection rate was 48.1% (62/129) among the HCWs. None of the HCWs were hospitalized and died.

The cumulative incidences in Puli Township and the nursing homes are shown in Figure 1. The total population of Puli Township was 77,587 in 2022. The COVID-19 epidemic in Puli Township started on 25 April 2022 [24], and the outbreaks in nursing homes began on 16 May 2022. The cumulative incidence in the three nursing homes is strongly associated with that in Puli Township (r = 0.978, *p* < 0.001), with a delay of approximately 21 days between the onset of the epidemic in the community and the outbreaks in the nursing homes. The cumulative incidences of the nursing home outbreaks and the epidemic in Puli Township reached 44.3% and 23.0%, respectively, at the end of the outbreak.

The numbers of cumulative cases in the three nursing homes are shown in Figure 2. The number of cumulative cases included both residents and HCWs. Three index cases occurred in our nursing homes: an HCW in Unit F on 16 May 2022, a resident in Unit B on 20 May 2022, and a resident in Unit P on 31 May 2022, respectively. The cumulative cases in the three nursing homes increased as mass screening tests were performed. The curve of cumulative cases in Unit P rose rapidly one week after the index case was identified (Figure 2).

The epidemic histogram, which displays the weekly count of new COVID-19 cases among the residents and HCWs in the nursing homes during the outbreak period, is plotted in Figure 3. The number of newly diagnosed cases in the three nursing homes reached its peak in the 3rd week of the outbreak, with some sporadic cases in the following 2 months and the last newly diagnosed case on 23 July 2022 (Figure 3).

### 3.3. Main Results

Cox regression was not performed for the infection rate of the HCWs because the majority of them were healthy without underlying diseases. The Cox regression model for significant risk factors associated with COVID-19 infection among the residents of the three nursing homes is listed in Table 5. All the variables were analyzed, but those without statistically significant differences are not listed. The univariate Cox regression model determined that the risk of COVID-19 infection increased in males (HR 2.12, 95% CI 1.27–3.52, *p* = 0.004), as well as severe dependency (HR 2.70, 95% CI 1.82–4.01, *p* < 0.001), other lower levels of dependency (i.e., Barthel index ≥ 61, including moderate dependency, mild dependency, and total independency) (HR 2.42, 95% CI 1.59–3.70, *p* < 0.001), and dementia (HR 1.46, 95% CI 1.04–2.05, *p* = 0.027).

The multivariate Cox regression model determined that the risk of COVID-19 infection increased in males (HR 2.46, 95% CI 1.47–4.11, *p* = 0.001), as well as severe dependency (HR 2.20, 95% CI 1.40–3.47, *p* = 0.001), lower levels of dependency (i.e., Barthel index ≥ 61, including moderate dependency, mild dependency, and total independency) (HR 1.93, 95% CI 1.18–3.17, *p* = 0.009), and dementia (HR 1.61, 95% CI 1.14–2.27, *p* = 0.007).

The statistically significant risk factors associated with COVID-19 death for the residents, determined via Cox regression, are listed in Table 6. All the variables were analyzed, but those without statistically significant differences are not listed. The univariate Cox regression model determined that risk of COVID-19 death increased with increasing age (HR 1.05, 95% CI 1.01–1.10, *p* = 0.026), consent to a DNR order (3.41, 95% CI 1.09–10.7, *p* = 0.036), pneumonia (HR 13.62, 95% CI 3.84–48.33, *p* < 0.001), hospitalization (HR 9.11, 95% CI 2.57–32.31, *p* = 0.001), and admission to ICU (HR 12.36, 95% CI 4.19–36.44, *p* < 0.001).

The age-adjusted multivariate Cox regression model determined that risk of COVID-19 death increased with consent to a DNR order (2.65, 95% CI 0.83–8.49, *p* = 0.1), pneumonia (HR 11.03, 95% CI 3.02–40.31, *p* < 0.001), hospitalization (HR 7.18, 95% CI 1.97–26.25, *p* = 0.003), and admission to ICU (HR 8.67, 95% CI 2.79–26.89, *p* < 0.001).

## 4. Discussion

Infection caused by the Omicron variant of SARS-CoV-2 has been reported to have lower rates of hospitalization and mortality in the general population, compared with the Delta variant [32,33], but information regarding outbreaks in nursing homes during the Omicron-predominant period is limited. The contribution of this study has been to evaluate the epidemic trend and risk factors associated with COVID-19 infection and related death among the residents and HCWs in nursing homes during the period of Omicron variant predominance.

This study reports that the rates of COVID-19 booster doses were 70.3% among the residents and 93.0% among HCWs before the outbreaks in the nursing homes. However, even during the outbreaks, the rates of COVID-19 infection remained high, reaching 41.6% among the residents and 48.1% among HCWs. A retrospective cohort study during the Alpha and Delta variants epidemic period found vaccine effectiveness against SARS-CoV-2 infection decreased from 81.3% (at month 2) to 57.8% (at month 8), losing about 5% of the vaccine effectiveness per month [34]. A study conducted before the Omicron predominant period showed that the SARS-CoV-2 vaccine’s effectiveness in residents of long-term care facilities declined from 12 weeks after a primary course of ChAdOx1-S or mRNA vaccines [35]. During the Omicron epidemic period, a booster dose of mRNA-1273, administered after either the BNT162b2 or ChA-dOx1 nCoV-19 primary vaccination course, substantially increased effectiveness to about 70% in protecting against symptomatic disease. However, protection waned over time, with the effectiveness falling to about 60% at 5 to 9 weeks [36]. All of these studies have consistently shown that vaccine effectiveness decreases over time. Despite a high vaccination rate, the infection rate during the outbreak remained notably high, suggesting a potential decline in vaccine effectiveness. In our study, we discovered that the intervals between a booster dose and the occurrence of COVID-19 infection were 101.8 ± 32.0 days for residents and 115.6 ± 48.7 days for HCWs. This finding raises the possibility that vaccine effectiveness may have waned over this relatively long period, contributing to the high infection rate observed in this study.

The psychiatric nursing home (Unit P) had the smallest dining area per resident dining together, with only 4.1 square meters compared to 30.4 square meters in Unit B and 32.4 square meters in Unit F. The infection rate in the psychiatric Unit P was higher than in the other two general units. Crowded nursing homes are more likely to experience larger COVID-19 outbreaks [23,37,38]. Close proximity among residents in small dining areas increases the risk of the virus spreading from person to person through respiratory droplets [38]. Residents with severe psychiatric symptoms may not follow even simple precautions, like hand washing or social distancing [18]. Psychiatric residents are at increased risk of COVID-19 infection as they interact closely in group therapy and use shared spaces, such as bathrooms and dining rooms. They are more likely to get infected with COVID-19 than the general population [39].

A number of asymptomatic individuals, including 54 (16.5%) residents and 15 (11.6%) HCWs, were detected through mass screening surveillance tests in our study. To interpret the COVID-19 infection rate and case fatality rate, we have to be aware that the increased detection of asymptomatic cases, as a confounding factor, has led to a reduction in these rates [40]. In a systematic review and meta-analysis, the pooled percentage of asymptomatic infections was 32.40% among SARS-CoV-2 Omicron variant-positive individuals [41]. The sensitivity of a rapid antigen test may reach up to 80.9% for symptomatic individuals and 64.3% for asymptomatic individuals [42], indicating that false negatives are also a confounding factor in assessing infection rates. In the early stages of the pandemic, there was an insufficient supply of testing kits to support surveillance testing. As testing capacity increases, COVID-19 prevalence and its related risk factors could change significantly [43].

The case fatality rate among residents of nursing homes is significantly higher than that of the general population. The global case fatality rate was 8.5% in February 2020, and it decreased to 0.27% in August 2022 [44]. The estimated relative risk reduction over 2.5 years was 96.8% (95% confidence interval [CI] 95.6–97.6, *p* < 0.001) [44]. However, the overall COVID-19 case fatality rate among infected residents remained high at 10.3% in our study. Additionally, the COVID-19-related mortality rate among all the residents in the three nursing homes was 4.3%. However, mortality rates among residents have shown variation in different studies. An analysis of pooled data from 14 countries across the USA and Europe until October 2021 revealed the proportion of COVID-19 deaths among occupied beds in nursing homes ranging from 2.4% in Germany to 7.7% in Spain and 9.8% in the USA [6].

In this study, a multivariate Cox regression model revealed that residents’ characteristics, such as not having total dependency or tube feeding, indicating higher levels of physical activity and contact with others, were associated with an increased risk of infection. In a cohort study of over 480,000 long-stay nursing home residents in the USA from 1 April 2020 to 30 September 2020, the risk of COVID-19 infection was primarily associated with geography and the specific nursing home facility, with minimal contribution of residents’ characteristics (including demographic variables and underlying diseases) [45]. A retrospective cohort study of 57 nursing homes in Italy included a total of 5145 residents from 1 March 2020 to 1 June 2020 [46]. Neither structural/organizational characteristics nor IPC measures were found to be significantly associated with changes in the prevalence of COVID-19 in nursing homes [46]. The lack of an association between IPC measures and the incidence of COVID-19 cases could be explained by the unmeasured IPC practices, such as the management difficulties, including lack of personal protective equipment, shortage of HCWs, an inability to have PCR or rapid antigen testing, and insufficiency of rooms for isolating COVID-19-infected residents in the first wave of the pandemic [46].

The Cox regression model used in this study revealed that increased age, pneumonia, hospitalization, and admission to the intensive care unit were associated with an increased risk of COVID-19 death. However, the study found that factors such as dependency level of ADL and underlying diseases were not associated with a higher risk of COVID-19 death among the residents. This lack of association could be attributed to the characteristics of the study population, which predominantly consisted of older individuals, males, and those with multiple chronic health conditions and physical function impairment.

Residents in long-term care facilities (LTCFs) are characterized by their old age compared with the general population. The median age among the LTCFs was 81 years in the USA and 87 years in Spain and Sweden [47]. The prevalence and severity of frailty are strongly associated with age [47]. The overall incidence of frailty was 50% among ambulatory individuals with a mean age of 81 and approximately 70% among residents in LTCFs with a mean age of 86 [47,48]. The incidence of COVID-19-related death is much higher in LTCFs than in the general population [47,49,50]. The risk factors for COVID-19-related death among the residents in LTCFs are different from those in the general population. Using a large hospital administrative database in the USA, the CDC assessed the in-hospital mortality risk for the general population by demographic and clinical characteristics during the Delta and Omicron period [50]. During the later Omicron period, most of the deaths happened among adults ≥ 65 years (81.9%) and patients with three or more underlying diseases (73.4%) [50]. Another large-scale study of 5256 residents among 351 USA nursing homes with COVID-19 outbreaks did not observe any significant association between 30-day mortality risk and certain chronic conditions (i.e., coronary artery disease, heart failure, chronic obstructive pulmonary disease, and hypertension), but they did find an association with chronic kidney disease and diabetes [49]. Mortality risk was associated more strongly with age and frailty (including physical and cognitive impairments), although their associations might at least partially reflect an increased risk of exposure to the underlying diseases [49]. While the presence of a single underlying disease may not be a reliable predictor of death, pneumonia poses a primary risk of COVID-19-related mortality in our study. Pneumonia symptoms in residents with impairments in physical and cognitive function can be vague and non-specific. Hence, it is important to closely monitor residents experiencing symptoms such as dyspnea, anorexia, falls, and/or altered consciousness. Early diagnosis and prompt treatment of pneumonia in vulnerable residents with subtle symptoms is essential to protect them from potential COVID-19-related death.

Although the infection rate (48.1%) of HCWs in the three nursing homes was high, none of the HCWs were hospitalized and died. In general, the overall infection rate of COVID-19 among HCWs is neither reported nor comparable across countries, and COVID-19 infections among HCWs may be correlated with the epidemic status in each country [15]. A higher nurse workforce may be associated with fewer COVID-19 cases, and facilities with nurse shortages are more susceptible to COVID-19 outbreaks [23,51]. To solve the problem of workforce shortage, we have implemented an effective policy by offering temporary and emergent support to strengthen the registered nurse workforce through the deployment of hospital staff. We also have provided accommodations for their temporary stay to prevent transmission between nursing homes and communities with a high prevalence of COVID-19, although some other authors consider that such measures are clearly not practical or sustainable in the vast majority of instances [47]. Our government also has offered an additional bonus incentive to HCWs caring for COVID-19 patients.

Researchers have found that residents and HCWs with positive test results were asymptomatic at the time of performing facility-wide surveillance tests [52,53,54], and these infected residents or HCWs may contribute to transmission even when asymptomatic [53]. Mass screening is particularly valuable for identifying asymptomatic or pre-symptomatic carriers of the virus, as well as symptomatic patients. Greater surveillance testing of staff members at nursing homes was associated with significant reductions in COVID-19 cases and deaths among residents, particularly before effective vaccines were available [19,20,21]. Although mass screening tests had started early on the first day of the index case being found, new cases were still occurring during the outbreaks in our study. There were several possibilities that contributed to the outbreak, including HCWs acquiring COVID-19 from their families or community, the environments being contaminated with the SARS-CoV-2 virus [23], and the residents and HCWs not being identified and isolated at the early stage of infection, with their viral load being too low to be detected by mass screening tests [42]. Although mass screening through regular surveillance testing may not entirely prevent outbreaks, we suggest it as an effective strategy for the early detection of COVID-19 and for preventing transmission during an epidemic period.

The cumulative incidence in the three nursing homes is highly linked to that in Puli Township, with a delay of approximately 21 days between the onset of the epidemic in the community and the outbreaks in the nursing homes. The positive correlation between the cumulative incidence of COVID-19 in the surrounding community and outbreaks in nursing homes is similar to the results of previous studies [45,46]. Another study also showed temporal associations between the incidence of COVID-19 in the community and the outbreaks in nursing homes, with an average lag time of 23 days [55]. During this window, several issues could be addressed to mitigate the impact of nursing home outbreaks. These include prioritizing the effective vaccination of nursing home residents and staff to reduce the severity of outbreaks and protect vulnerable populations. Another is implementing mass screening through regular testing for residents and staff to quickly identify and isolate positive cases in order to prevent the introduction and spread of the virus in nursing homes [19,20,21,43]. Additionally, nursing homes may limit or reduce the frequency, duration, and volume of visitations and group activities to minimize the risk of virus transmission within the facility [5,23].

There were limitations in this study. Firstly, the homogeneity of the population: Nursing home populations consist mainly of elderly individuals with comorbidities, resulting in a relatively homogeneous study population. It also limits our ability to explore the impact of COVID-19 on diverse groups. Secondly, the exact routes of transmission among the residents and staff could be multifactorial; HCWs who are exposed to the virus in the community may subsequently transmit it to residents and other HCWs during their work shifts. Asymptomatic carriers infected with COVID-19 could unknowingly spread the virus. Additionally, shared facilities, such as dining rooms, activity rooms, and therapy rooms, could facilitate virus transmission when infected individuals are present. The virus could also be transmitted through contaminated surfaces. These various routes of transmission could significantly impact the statistical analysis of risk factors for COVID-19 infections, especially underlying diseases. Thirdly, some of the unmeasured infection prevention practices may act as confounding factors. The risk of infection may increase due to inappropriate IPC measures, such as improper mask-wearing, inadequate hand hygiene, and failure to maintain social distancing. However, the method used to observe and evaluate these IPC measures can also introduce bias, as individuals in a nursing home may occasionally inadvertently or unknowingly violate IPC measures without staff noticing and without records.

## 5. Conclusions

This study highlights the high infection rate among residents and HCWs, as well as a high case fatality rate among the residents in nursing homes during the Omicron epidemic period. Vaccine effectiveness may have declined over time, contributing to the high infection rate. This emphasizes the importance of promoting and ensuring higher vaccination rates with timely administration of effective next-generation booster vaccines to fight against new variants of concern and prevent the waning vaccine effectiveness.

Given the substantial rates of asymptomatic infections among both residents and HCWs, relying solely on traditional symptom monitoring may not be sufficient for detecting infections. We suggest implementing mass screening through regular surveillance testing as an effective strategy for early COVID-19 detection and for preventing transmission during an epidemic period.

This study identified specific risk factors for COVID-19 infection and mortality among nursing home residents using the Cox regression model. Residents who had lower dependency levels and engaged in more group activities were found to have an increased risk of COVID-19 infection. This suggests that group activities in shared spaces may increase the risk of virus transmission among residents, highlighting the importance of considering restrictions or reductions in such activities and improving compliance with IPC measures. This study found that factors such as the level of dependency on ADL and underlying diseases were not associated with a higher risk of COVID-19 death among the residents. However, pneumonia, hospitalization, and admission to the ICU were found to be associated with an increased risk of COVID-19 death. Pneumonia symptoms in residents can be vague and non-specific. Early detection and prompt treatment of COVID-19 pneumonia for vulnerable residents with subtle symptoms are crucial to protect them from potential mortality.

## Figures and Tables

**Figure 1 healthcare-11-02868-f001:**
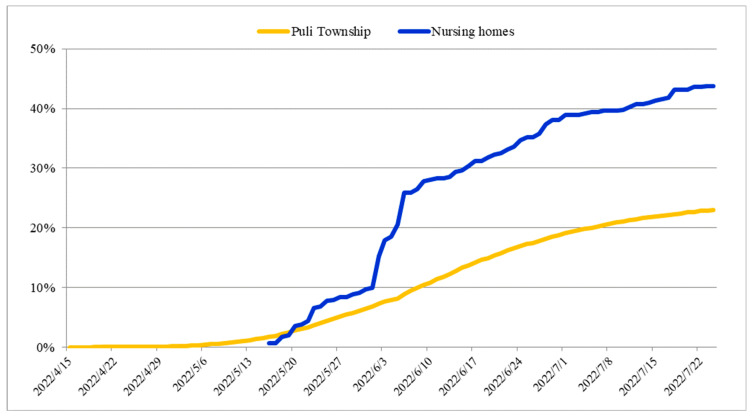
Cumulative incidence of COVID-19 in Puli Township and the three nursing homes. The *x*-axis represents the time period during which the disease outbreak occurred. The *y*-axis represents the cumulative incidence of COVID-19 reported. The cumulative incidence in the three nursing homes includes the data of both residents and HCWs. The blue curve represents the cumulative incidence of all three nursing homes, and the orange curve represents that of Puli Township.

**Figure 2 healthcare-11-02868-f002:**
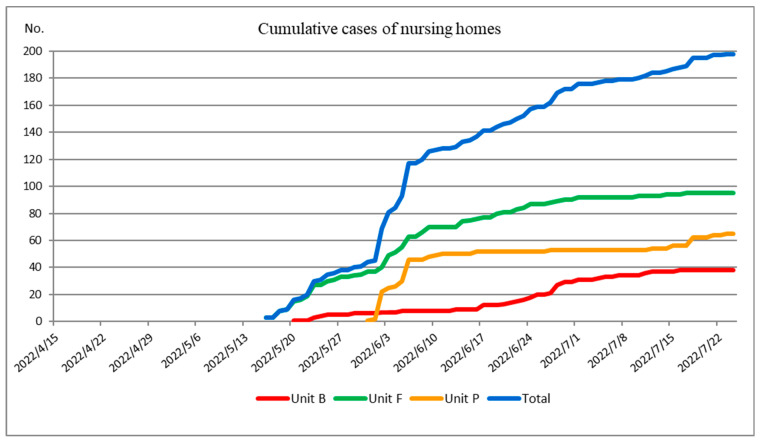
Cumulative cases of COVID-19 in the three nursing homes. The *x*-axis of the curve represents the time period during which the outbreak occurred. The *y*-axis represents the number of cumulative cases of COVID-19. The red curve represents the cumulative cases of Unit B, the green curve represents the cumulative cases of Unit F, the orange curve represents the cumulative cases of Unit P, and the blue curve represents the total cumulative cases of the three units.

**Figure 3 healthcare-11-02868-f003:**
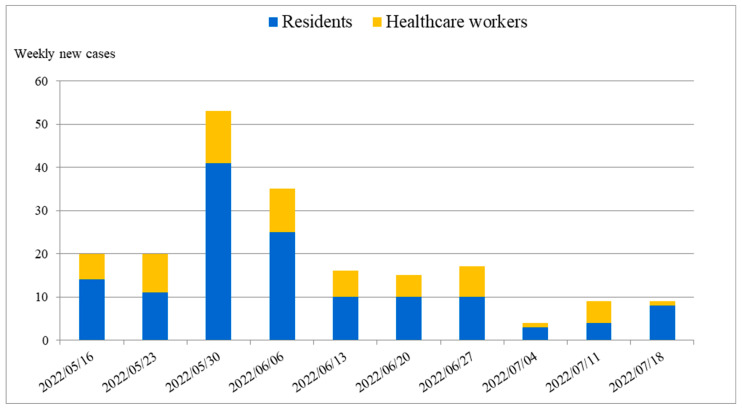
Epidemic histogram of weekly new COVID-19 cases among both the residents and healthcare workers in the nursing homes during the outbreak period. The *x*-axis represents the time period during which the outbreak occurred. The *y*-axis represents the number of weekly new cases. The blue bars represent the number of weekly new cases among the residents, while the orange bars represent the number of weekly new cases among the healthcare workers.

**Table 1 healthcare-11-02868-t001:** The building space of the three nursing homes.

	Unit B	Unit F	Unit P	Total
**Space for all residents**				
Number of residents	87	184	56	327
Total space (m^2^)	3289.6	5485.5	2374.7	11,149.8
Average space (m^2^) per resident	37.8	29.8	42.4	34.1
**Space for residents dining together**				
No. of residents dining together	11	22	56	89
Space of public dining area (m^2^)	334.7	712.7	230.9	1278.2
Average public dining space (m^2^) per resident	30.4	32.4	4.1	14.4

**Table 2 healthcare-11-02868-t002:** Demographics and underlying diseases of the residents in the three nursing homes.

	Unit B *N* = 87	Unit F *n* = 184	Unit P *n* = 56	Total *n* = 327	*p*
**Gender**					0.023
Male	78 (89.7%)	142 (77.2%)	41 (73.2%)	261 (79.8%)	
Female	9 (10.3%)	42 (22.8%)	15 (26.8%)	66 (20.2%)	
**Age (mean ± SD)**	76.5 ± 17.3	77.5 ± 15.3	74.5 ± 14.9	76.2 ± 15.4	0.587
**Consent to a DNR order**	52 (59.8%)	110 (59.8%)	9 (16.1%)	171 (52.3%)	<0.001
**Medical supports**					
Tube feeding	57 (65.5%)	81 (44%)	0 (0%)	138 (42.2%)	<0.001
Artificial airway	10 (11.5%)	4 (2.2%)	0 (0%)	14 (4.3%)	<0.001
**Dependency of ADL**					<0.001
Total dependency	68 (78.2%)	130 (70.7%)	5 (8.9%)	203 (62.1%)	<0.001
Severe dependency	10 (11.5%)	34 (18.5%)	23 (41.1%)	67 (20.5%)	<0.001
Moderate dependency	5 (5.7%)	15 (8.2%)	18 (32.1%)	38 (11.6%)	<0.001
Mild dependency	2 (2.3%)	4 (2.2%)	5 (8.9%)	11 (3.4%)	NA
Independent	2 (2.3%)	1 (0.5%)	5 (8.9%)	8 (2.4%)	NA
**Underlying diseases**					
Hypertension	47 (54%)	97 (52.7%)	33 (58.9%)	177 (54.1%)	0.716
Dementia	33 (37.9%)	63 (34.2%)	41 (73.2%)	137 (41.9%)	<0.001
Cerebral vascular disease	43 (49.4%)	74 (40.2%)	19 (33.9%)	136 (41.6%)	0.158
Diabetes mellitus	38 (43.7%)	56 (30.4%)	15 (26.8%)	109 (33.3%)	0.051
Chronic kidney disease	22 (25.3%)	31 (16.8%)	11 (19.6%)	64 (19.6%)	0.263
Chronic pulmonary disease	16 (18.4%)	30 (16.3%)	11 (19.6%)	57 (17.4%)	0.815
Heart failure	17 (19.5%)	23 (12.5%)	4 (7.1%)	44 (13.5%)	0.090
Hyperlipidemia	14 (16.1%)	19 (10.3%)	8 (14.3%)	41 (12.5%)	0.372
Coronary artery disease	17 (19.5%)	20 (10.9%)	2 (3.6%)	39 (11.9%)	0.012
Advanced malignancy	6 (6.9%)	15 (8.2%)	3 (5.4%)	24 (7.3%)	0.878
End-stage renal disease	4 (4.6%)	6 (3.3%)	2 (3.6%)	12 (3.7%)	0.919
Chronic liver disease	0 (0%)	5 (2.7%)	2 (3.6%)	7 (2.1%)	0.235

NA: not applicable due to small sample size.

**Table 3 healthcare-11-02868-t003:** Demographic characteristics and underlying diseases among the 101 healthcare workers obtained via questionnaire.

	Registered Nurses (*n* = 28)	Nursing Assistants (*n* = 59)	Social Workers (*n* = 3)	Housekeepers and Porters (*n* = 7)	Clerks (*n* = 4)	Total (*n* = 101)
**Gender**						
Female	27 (96.4%)	50 (84.7%)	1 (33.3%)	3 (42.9%)	4 (100%)	85 (84.2%)
Male	1 (3.6)	9 (15.3%)	2 (66.7%)	4 (57.1%)	0 (0%)	16 (15.8%)
**Age**	39.2 ± 9.2	42.5 ± 12.6	40.3 ± 12.7	58 ± 5.6	34 ± 3.9	42.3 ± 11.9
**Underlying diseases**						
Diabetes mellitus	1 (3.6%)	4 (6.8%)	0 (0%)	2 (28.6%)	0 (0%)	7 (6.9%)
Hypertension	3 (10.7%)	11 (18.6%)	0 (0%)	2 (28.6%)	0 (0%)	16 (15.8%)
Chronic liver disease	0	2	0	0	0	2 (2.0%)
Cancer	1 (3.6%)	0	0	0	0	1 (1%)
Congestive heart failure	0	0	0	0	0	0 (0%)
COPD	0	1	0	0	0	1 (1%)
Hyperlipidemia	NA	NA	NA	NA	NA	NA
Chronic kidney disease	NA	NA	NA	NA	NA	NA

NA: not available.

**Table 4 healthcare-11-02868-t004:** Vaccination- and COVID-19-related statistics among the residents and healthcare workers in the three nursing homes.

	Unit B	Unit F	Unit P	Total	*p*
**Residents**					
Number of residents	87	184	56	327	
Vaccination rate	53 (60.9%)	127 (69.0%)	50 (89.3%)	230 (70.3%)	0.001
Average days between a booster dose and COVID-19 infection	80.9 ± 29.1	97.7 ± 26.1	134.5 ± 22.8	101.8 ± 32.0	NA
Asymptomatic residents with positive COVID-19 test	7 (8.0%)	25 (13.6%)	22 (39.3%)	54 (16.5%)	<0.001
Infection rate	22 (25.3%)	63 (34.2%)	51 (91.1%)	136 (41.6%)	<0.001
Hospitalization	13 (59.1%)	18 (28.6%)	13 (25.5%)	44 (32.4%)	0.013
Case fatality rate	2 (9.1%)	9 (14.3%)	3 (5.9%)	14 (10.3%)	0.355
COVID-19-related mortality rate	2 (2.3%)	9 (4.9%)	3 (5.4%)	14 (4.3%)	0.567
**Healthcare workers**					
Number of HCWs	38	66	25	129	
Vaccination rate	36 (94.7%)	61 (92.4%)	23 (92.0%)	120 (93.0%)	1.000
Average days between a booster dose and COVID-19 infection	110.5 ± 35.6	114.8 ± 61.6	125.7 ± 18.3	115.6 ± 48.7	NA
Asymptomatic HCWs with positive COVID-19 test	1 (2.6%)	7 (10.6%)	7 (28.0%)	15 (11.6%)	0.01
Infection rate	16 (42.1%)	32 (48.5%)	14 (56.0%)	62 (48.1%)	0.549
Hospitalization	0 (0%)	0 (0%)	0 (0%)	0 (0%)	0 (0%)
COVID-19-related mortality rate	0 (%)	0 (0%)	0 (0%)	0 (0%)	0 (0%)

NA: not applicable.

**Table 5 healthcare-11-02868-t005:** Cox regression model for significant risk factors associated with COVID-19 infection among the residents of the three nursing homes.

	Infected *n* = 136	Not Infected *n* = 191	Univariate HR (95% CI)	*p* Value	Multivariate HR (95% CI)	*p* Value
**Age (mean ± SD)**	76.2 ± 15.4	79.7 ± 13.8	0.99 (0.98–1.00)	0.056	0.99 (0.98–1.00)	0.149
**Gender**						
Male	119 (87.5%)	142 (74.3%)	2.12 (1.27–3.52)	0.004	2.46 (1.47–4.11)	0.001
Female	17 (12.5%)	49 (25.7%)	1.00 (reference)		1.00 (reference)	
**Dependency of ADL**						
Total dependency	59 (43.4%)	144 (75.4%)	1.00 (reference)		1.00 (reference)	
Severe dependency	43 (31.6%)	24 (12.6%)	2.70 (1.82–4.01)	<0.001	2.20 (1.40–3.47)	0.001
Other dependencies	34 (25.0%)	23 (12.0%)	2.42 (1.59–3.70)	<0.001	1.93 (1.18–3.17)	0.009
**Tube feeding**						
No	98 (72.1%)	92 (48.2%)	1.00 (reference)		1.00 (reference)	
Yes	38 (27.9%)	99 (51.8%)	0.48 (0.33–0.69)	<0.001	0.65 (0.41–1.02)	0.063
**Dementia**						
No	69 (50.7%)	121 (63.4%)	1.00 (reference)		1.00 (reference)	
Yes	67 (49.3%)	70 (36.6%)	1.46 (1.04–2.05)	0.027	1.61 (1.14–2.27)	0.007

“Other dependencies” indicates those levels with Barthel index ≥ 61, including moderate dependency, mild dependency, and total independency.

**Table 6 healthcare-11-02868-t006:** Cox regression model for risk factors associated with COVID-19 death among the residents of the three nursing homes.

	Death *n* = 15	Survival *n* = 121	Univariate HR (95% CI)	*p* Value	Multivariate Adjusted-for-Age HR (95% CI)	*p* Value
**Age (mean ± SD)**	84.9 ± 11.2	75.1 ± 15.6	1.05 (1.01–1.10)	0.026		
**Consent to a DNR order**						
No	4 (26.7%)	70 (57.9%)	1 (reference)		1 (reference)	
Yes	11 (73.3%)	51 (42.1%)	3.41 (1.09–10.71)	0.036	2.65 (0.83–8.49)	0.100
**Pneumonia**						
No	3 (20.0%)	98 (81.0%)	1 (reference)		1 (reference)	
Yes	12 (80.0%)	23 (19.0%)	13.62 (3.84–48.33)	<0.001	11.03 (3.02–40.31)	<0.001
**Hospitalization**						
No	3 (20.0%)	88 (72.7%)	1 (reference)		1 (reference)	
Yes	12 (80.0%)	33 (27.3%)	9.11 (2.57–32.31)	0.001	7.18 (1.97–26.25)	0.003
**Intensive care**						
No	10 (66.7%)	120 (99.2%)	1 (reference)		1 (reference)	
Yes	5 (33.3%)	1 (0.8%)	12.36 (4.19–36.44)	<0.001	8.67 (2.79–26.89)	<0.001

## Data Availability

Data will be made available upon request to the corresponding author.

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
