# Peer review of "The Epidemiological Analysis of COVID-19 Outbreaks in Nursing Homes during the Period of Omicron Variant Predominance"

_healthcare, 2023, doi:10.3390/healthcare11212868_

Round 1

Reviewer 1 Report

Dear Authors,

Thank you for your interesting topic. I would like to suggest improvement on the following:

Introduction

- I suggest for you to include in your introduction the background of your study specifically the context in the three affiliated nursing homes of Taichung veterans hospital.

- You included the HCWs in the survey but you did not mention their surrounding circumstance in the nursing homes.

-Aside from the vulnerability of the residents by virtue of their age and to some the presence of existing chronic diseases or comorbidities which are known to increase their risk for Covid-19 morbidity and mortality, I suggest that the preventive measures other than vaccination and distancing should be included. These can be considered as factors for the outbreak if any.

-  What is the specific gap in these  3 nursing homes that led  or mtivated you to do the study? The statement of the aim of the study should include the specific details such as which  nursing homes, the population (whether you focus on the residents or HCWs or both of them). The thesis statement should be clearly discussed in this section.

Methods

I suggest that this section be strengthened. Include subsections for the study design, the instruments used (including the validity and reliability), population of the study, etc

subsection 2.1 - include other preventive measures other than vaccination and distancing for both residents and hcws (example: biomonitoring) These are important in determining confounding variables or other factors to the covid-19 related morbidity and mortality in the nursing homes.

Design - identify what type of epidemiological study , the specific variables and outcomes measured in the study, confounding variables and how these were controlled. You only mentioned infection and death as your outcome, kindly specify or define if these include only covid-19 related mortality and morbidity

Population - describe in separate paragraph the residents and the CHWs (sample size, inclusion criteria, etc)

Instrument- Describe the tools/instruments you used to gather the information needed to answer your research aim/objective. Include the validity and reliability.

Describe any biases if any.

Statistical analysis- include your descriptive statistics on how you interpret, analyzed the result of the surveys (eg. demographics, underlying diseases)

Results;

Present two sets of results (residents and chws all throughout)

The variables included in the study are expected risk factors among this specific vulnerable groups. It would add more value if other factors were included in the study.

Categorize the presentation of your results.

Discussion- Support your discussion with more references. Add more depth and breadth in the analysis of your results.

Conclusion- How about the chws?The conclusion should not be a summary of the results but should include your inference in the light of the findings.

Overall - clarify from the start if the study focused only on the residents orthe chws are included. If the chws are included, outline  separately the variables and outcomes unique for each group and make sure these two groups are well covered all throughout the paper from the introduction to the conclusion.

Thank you.

English editing should be considered.

Reviewer 2 Report

Thank you for inviting me to review this paper. Below are my comments.

Introduction

Lines 46 to 48: The sentence infers that there is no basis for this study. The authors should provide a reference for the statement.

Lines 48 to 50: The sentence contradicts the previous sentence in lines 46 to 48. The authors should provide a reference for the statement.

Lines 66 & 67: This study did not address the identified first limitation of the previous study.

Methods

The authors should rewrite the method section following the STROBE guideline for clarity. All the variables were not clearly identified and placed in the appropriate section. The authors did not mention how the data was collected. There was no clear definition of the participants. Who were the participants? Residents? HCWs? Both? Following the guidelines will ensure that all the necessary information is captured.

Line 83: Change every number less than ten to letters.

Line 85: The word “another” can be replaced by “the third”. Ensure it reads well.

Lines 89 to 96: I guess the paragraph is about the strategies of infection prevention and control. If true, then it is only the first sentence that is suited. Moreover, the entire paragraph is not needed in the method section. The strategies of infection prevention and control should be infused into the introduction.

Line 100: The date format should be changed. The appropriate format is “May 16, 2022” or “16th May 2022.” Kindly effect the correction throughout the manuscript.

Line 107: The sentence should have started the paragraph in line 99. What informed the stopping of the observation on August 14, 2022?

Lines 108 & 109: Was each unit compared with the Puli Township? Where and how did the authors collect the data about the incidence of COVID-19 in the town? How did the authors collect other data in this study? The cumulative incidence should be moved to the variable subsection.

Line 111: Many variables were not described in the study variables. The authors should define all the variables and show outcome and predictor variables.

Line 119: The definition of the vaccination rate is not quite clear. Must they receive a booster dose? What do the authors mean by "more than 14 days?"

Statistical analysis

The authors should discuss the descriptive and the inferential analysis.

Lines 132 to 143: The definition of those variables should be moved to the variable subsection.

Lines 135 to 138: This is not shown in the result. First, no curve was plotted; it was a bar chart. Second, it did not depict duration and transmission patterns. The bar chart shows the number/magnitude of new cases per week. The authors should define the variables clearly, state the analysis, and present their results in line with the analysis.

Lines 138 & 139: What is the difference between cumulative incidences and cumulative cases? Kindly define your variables appropriately.

Lines 144 & 145: While it is good to conduct Chi-square and ANOVA tests, it is better to complete pairwise comparisons for significant variables. What do the authors mean by categorical and continuous features? I guess they mean variables. If true, then what are those categorical and continuous variables?

 Results

Lines 185 to 187: Do you work in the nursing homes? How did you know the status of the index cases? Mentioning a nursing aide is an ethical breach. It will be better to remove the status of the index cases in the report. Moreover, if HCWs are participants in this study, then their descriptive statistics should be provided.

Line 198 (Table 1): Why did the authors not compare the differences for some variables as done for others? The table footnotes should contain the meaning of abbreviations and other symbols used in the table. It should not contain the meaning of variables.

Line 219: Figure 3 is not a curve. It is bar chat.

Line 224: Who are the participants? The residents? The HCWs? Both? Were the demographics completed before the outbreak? The demographic information should come first in the result section.

Line 227: The authors should be consistent in their reporting. It is better to stick to the figures in the table rather than rounding them up. The 90% is not in the table, rather it is 89.7%.

Line 238 (Table 2): Bolden the variables to differentiate them from the categories. Why were other categories and variables (such as consent to DNR, pneumonia, tube feeding, and intensive care) not included in the descriptive analysis?

Lines 250 & 264: What determined the variables included in the multivariate analysis model? Why are the variables different for each of the multivariate models? Is the model significant?

Line 264: How did the authors adjust for age?

Line 270 (Table 3): Bolden the variables to differentiate them from the categories. The authors should indicate the reference category for each variable. Are female and total dependency the reference for gender and dependency of ADL, respectively? I guess the answer is yes.

Discussion

The discussion should flow sequentially with the results. The authors’ discussion deviated from the results

Lines 287 to 306: This is like comparing an apple with an orange. Discuss why the Omicron outbreak was high despite the high vaccination rate. The study is not about the effectiveness of the vaccines.

Line 307: The information is not in the result. The study is not about the effectiveness of the vaccines.

Line 312: The information is not in the result.

Line 325: Is “total dependency” not the reference category in the model? Why include it in the discussion?

This is not too bad at all but can be improved. See my comments above.

Round 2

Reviewer 1 Report

Dear authors,

Thank you for the improved version of your paper and for your efforts to incorporate suggestions.

I have just few more notes for you to work on:

1. Study design - Kindly decide on the design of your study between cohort and cross-sectional design. 

2. Population - I suggest that the two populations be discussed in separate paragraphs. One paraghraph for the residents and one for the HCWs.

3. The conclusion should not be a summary of results but based on the findings what did you infer?

Thank you and good luck.

Very minor edition that the authors can do by themselves if they are confident to do so.
